# Astragaloside IV Ameliorates Cerebral Ischemic-Reperfusion Injury via Improving Mitochondrial Function and Inhibiting Neuronal Apoptosis

**DOI:** 10.3390/cimb47080597

**Published:** 2025-07-29

**Authors:** Tongtong He, Xiaohong Zhou, Xiaorong Wang, Yanmeng Zhao, Zhenyi Liu, Ping Gao, Weijuan Gao, Xiaofei Jin

**Affiliations:** Hebei Key Laboratory of Chinese Medicine Research on Cardio-Cerebrovascular Disease, Hebei University of Chinese Medicine, No. 3 Xingyuan Road, Luquan District, Shijiazhuang 050200, China; halona0211@hebcm.edu.cn (T.H.); zhouxiaohong@hebcm.edu.cn (X.Z.); yjs20232051@hebcm.edu.cn (X.W.); jiushihenhaoji@126.com (Z.L.); gaoping0321@163.com (P.G.)

**Keywords:** astragaloside IV, cerebral ischemic reperfusion injury, PINK1/Parkin pathway, reactive oxygen species

## Abstract

Cerebral ischemic-reperfusion injury (CIRI) involves mitochondrial dysfunction, with mitophagy playing a key role. Astragaloside IV (AS-IV) shows neuroprotective potential; however, its mechanisms related to mitochondrial function and apoptosis remain unclear. Methods: Using a rat MCAO/R model, we evaluated the AS-IV’s effects via neurological scores, TTC staining, and histopathology. Molecular assays and docking were used to analyze mitophagy (PINK1, Parkin, p62, ROS, Bcl-2, and BAX) and apoptosis markers. Results: AS-IV improved neurological function, reduced infarct volume, and alleviated neuronal/mitochondrial damage. It upregulated PINK1/Parkin, decreased p62, and modulated Bcl-2/Bax. Docking confirmed AS-IV binds PINK1/Parkin with high affinity. Conclusions: AS-IV protects against CIRI by regulating the PINK1/Parkin pathway, improving mitochondrial function, and inhibiting neuronal apoptosis, providing an experimental basis for the clinical use

## 1. Introduction

Cerebral ischemic-reperfusion injury (CIRI), as the core link of subsequent pathological damage in ischemic stroke, involves multiple complex pathological processes, such as mitophagy, oxidative stress, neuronal apoptosis, inflammatory response, and blood-brain barrier disruption [1,2,3]. Although current thrombolytic therapies (e.g., recombinant tissue plasminogen activator rt-PA) and neuroprotective strategies have made certain progress in restoring blood perfusion, clinical treatment still faces enormous challenges due to reperfusion-induced secondary injury, narrow therapeutic time window, and limited neurorepair effects [4]. Therefore, exploring the endogenous protective mechanisms of CIRI and identifying therapeutic approaches that can effectively alleviate reperfusion damage have become critical issues to be addressed in the field of neuroscience.

The occurrence of CIRI is closely related to oxidative damage and cell apoptosis caused by mitochondrial impairment [5]. As the “powerhouse of the cell,” mitochondria not only produce ATP to provide energy for the organism but also play important roles in regulating reactive oxygen species (ROS) production, maintaining calcium homeostasis, adjusting osmotic pressure, and transducing cellular signals [6,7]. Under physiological conditions, an antioxidant enzyme defense system exists in mitochondria (including superoxide dismutase (SOD), glutathione peroxidase (GSH-Px), etc.), which is crucial for maintaining the balance between ROS production and clearance [8]. Cerebral ischemia directly leads to a sharp reduction in intracerebral energy supply, which can cause damage to mitochondrial morphology, structure, and function, further exacerbating cellular energy production dysfunction [9]. During reperfusion, excessive ROS is generated and cannot be cleared promptly. When ROS accumulates excessively beyond the capacity of the mitochondrial antioxidant enzyme defense system, it triggers oxidative damage, simultaneously causing a decrease in mitochondrial membrane potential and opening of the mitochondrial permeability transition pore (MPTP) [10]. This leads to the release of pro-apoptotic substances into the cytoplasm, forming apoptotic protease complexes (i.e., apoptotic bodies), which further activate downstream Caspase-3, induce mitochondrial pathway apoptosis, and ultimately result in the CIRI [11].

Astragaloside IV (AS-IV), the main active monomer component of Astragalus membranaceus, exhibits extensive pharmacological activities [12] and shows great potential in the treatment of cardiovascular and neurological diseases [13,14]. Previous studies have demonstrated that AS-IV can alleviate CIRI by scavenging ROS and inhibiting inflammatory responses [15,16]; however, its regulatory effect on mitophagy and underlying molecular mechanisms remain unclear. Notably, other compounds have been reported to reduce CIRI by enhancing mitophagy; for instance, a recent study showed that a natural flavonoid derivative ameliorates CIRI in rats by activating the PINK1/Parkin-mediated mitophagy pathway, accompanied by reduced mitochondrial dysfunction and neuronal apoptosis [17]. Another study revealed that a diterpenoid compound protects against CIRI by upregulating mitophagy-related proteins (e.g., LC3-II/LC3-I, Beclin-1) and improving mitochondrial quality control [18]. These findings highlight the importance of the regulation of the PINK1/Parkin pathway in CIRI therapy, providing a rationale for exploring the role of AS-IV in this process.

Mitophagy is a process by which cells selectively clear damaged mitochondria through autophagic regulatory mechanisms, playing an important role in controlling mitochondrial quality and maintaining cell survival [19,20]. Currently, multiple signaling pathways, such as PTEN-induced kinase 1 (PINK1), E3 ubiquitin ligase Parkin, and PI3K-AKT, have been identified to be involved in mitophagy in mammals [21,22], among which PINK1 and Parkin play key regulatory roles in the initiation and progression of mitophagy [23]. Under physiological conditions, PINK1 enters mitochondria and is rapidly cleaved and degraded in the inner membrane, maintaining a stable, low-level state. During cerebral ischemia-reperfusion, excessive ROS production leads to oxidative imbalance in mitochondria, exacerbating mitochondrial damage, causing a decrease in mitochondrial membrane potential and the opening of MPTP. PINK1 can rapidly sense the decline in membrane potential, stably accumulate on the outer membrane surface to function as a biosensor, and induce Parkin translocation to damaged mitochondria, thereby triggering mitophagy [24,25]. Ubiquitinated, damaged mitochondria are wrapped by double-membraned autophagic vesicles to form mitophagosomes, which then gradually fuse with lysosomes to form mitophagolysosomes. The enclosed damaged mitochondria are degraded and utilized by lysosomal enzymes, thereby maintaining intracellular homeostasis [26]. Thus, the ROS–PINK1/Parkin signaling pathway plays an important role in mediating mitophagy and clearing damaged mitochondria.

Based on this, this study established a rat model of middle cerebral artery occlusion/reperfusion (MCAO/R)-induced CIRI to systematically investigate the protective effect of AS-IV on ischemic-reperfusion brain injury and its molecular mechanism in mediating mitophagy through the ROS-PINK1/Parkin pathway, aiming to provide experimental and theoretical support for its clinical application in the treatment of ischemic stroke.

## 2. Materials and Methods

### 2.1. Animals

Sixty SPF-grade male Sprague-Dawley (SD) rats, weighing 270–290 g, were provided by Beijing Vital River Laboratory Animal Technology Co., Ltd. (Beijing, China) Animal License No.: SCXK (Jing) 2016-0011. The experimental protocol was approved by the Ethics Committee of Hebei University of Chinese Medicine (Approval No.: DWLL202302022), and all operations strictly adhered to the guidelines for experimental animal ethics.

### 2.2. Main Reagents

AS-IV was purchased from Shifeng Biotechnology Co., Ltd. (Shanghai, China), with specifications of 20 mg per vial and purity ≥ 98% (Batch No. 15082136); Nimodipine Injection (H20033549) was obtained from Shandong Fangming Pharmaceutical Group Co., Ltd. (Jinan, China); FCCP (HY-100410) was obtained from MedChemExpress (Princeton, NJ, USA); Antibodies against PINK1 (23274-1-AP) and Parkin (14060-1-AP) were purchased from Wuhan Sanying Biotechnology Co., Ltd. (Wuhan, China); Antibodies against BAX (ab32503), Bcl-2 (ab196495), and p62 (ab109012) were obtained from Abcam (Cambridge, UK); TTC staining solution (G1017), phosphate-buffered saline (G4202), HE staining kit (G1005), mouse anti-β-actin monoclonal antibody (GB15001), rabbit anti-β-actin polyclonal antibody (GB15003), Nissl staining kit (G1036), and universal tissue fixative (G1101) were all purchased from Servicebio Technology Co., Ltd. (Wuhan, China); QuickBlock™ Immunostaining Blocking Buffer (P0260), DAPI-containing anti-fluorescence quenching mounting medium (P0131), SDS-PAGE gel preparation kit (P0012AC), and RIPA lysis buffer (P0013B) were obtained from Beyotime Biotechnology Co., Ltd. (Shanghai, China); Mitochondrial extraction kit was purchased from Sunshine Biotechnology Co., Ltd. (Beijing, China); Kits for detecting ATP (A095-1-1) and ROS (E004-1-1) levels were obtained from Nanjing Jiancheng Bioengineering Institute (Nanjing, China); Kits for detecting mitochondrial permeability transition pore (mPTP, GMS10101) and mitochondrial membrane potential (MMP, GMS10013.1) were purchased from Jinmei Gene Pharmaceutical Technology Co., Ltd. (Shanghai, China). Middle cerebral artery occlusion suture (2636-A3) was purchased from Xinnong Biotechnology Co., Ltd. (Beijing, China).

Equipment included a multi-functional small animal operating table (Kent, Washington, DC, USA), an electrophoresis system and semi-dry transfer membrane system (Bio-Rad, Hercules, CA, USA), a high-speed low-temperature tissue grinder (Servicebio Technology Co., Ltd., Wuhan, China), a refrigerated high-speed centrifuge, and a multi-functional microplate reader (Thermo, Waltham, MA, USA).

### 2.3. Methods

#### 2.3.1. Animal Grouping and Modeling

The rats were randomly divided into two sets of groups: (1) sham operation (Sham) group, model (MCAO/R) group, astragaloside IV (AS-IV) group, and nimodipine (NIM) group; (2) sham operation (Sham) group, model (MCAO/R) group, astragaloside IV (AS-IV) group, and Carbonyl cyanide-4-(trifluoromethoxy) phenylhydrazone (FCCP) group. Each group contained 15 rats. A rat model of focal CIRI was established using the middle cerebral artery occlusion/reperfusion (MCAO/R) method. Rats were adaptively fed for 1 week, fasted for 12 h, and deprived of water for 4 h before surgery. Rats were anesthetized by intraperitoneal injection of 2% pentobarbital sodium (40 mg/kg), fixed on a small animal operating table, and carefully prepared with skin disinfection using alcohol cotton balls. A small incision was made in the midline of the rat’s neck, and tissues were separated layer by layer to expose the left common carotid artery, external carotid artery, and internal carotid artery. The external carotid artery was ligated and blocked, and a specific suture was inserted through the stump of the external carotid artery, then advanced through the internal carotid artery to the rats’ middle cerebral artery to achieve ischemic occlusion. During the operation, the rats were placed on a constant temperature electric heating pad to maintain their core body temperature at (37.0 ± 0.5) °C. The suture was removed after 2 h of ischemia, followed by 24 h of reperfusion. Rats in the Sham group underwent the same surgical procedures except for suture insertion. Rats in the AS-IV group (20 mg/kg), the NIM group (10 mg/kg), and the FCCP (0.84 mg/kg) group were administered intraperitoneally at the start of reperfusion, and samples were collected after 24 h. It should be noted that some rats died during modeling; only successfully modeled rats were included in the experiment, with a modeling success rate of over 80%. Each group was ensured to have a sample size of no less than 15 rats (*n* ≥ 15 per group).

#### 2.3.2. Operational Method of Laser Speckle Blood Flow Imaging Monitoring Cerebral Blood Flow in Rat MCAO Model

After anesthetizing the rats, a longitudinal incision was made between the eyes and ears. The incision was secured with a hook to expose the cerebral cortex, with symmetrical regions of both cerebral hemispheres were selected as monitoring areas. The laser speckle flow imaging system lens (PERIMED, Stockholm, Sweden) was positioned at a fixed distance of 11.0 cm from the observation area. Blood flow data were collected three times: before the MCAO surgery (baseline state), two hours after single filament insertion (ischemic state), and after removal of the single filament (reperfusion state). By dynamically monitoring and comparing cerebral blood flow changes across different stages, this approach provides quantitative evidence for evaluating the model’s validity and intervention efficacy.

#### 2.3.3. Evaluation of Neurological Deficits Using Zea Longa Score

The Zea Longa scoring system was used for preliminary screening of neurological function. Rats with scores of 1–3 were selected and randomly assigned to each experimental group, while those with a score of 0 (no neurological deficit) or 4 (moribund state) were excluded. The MCAO/R model was established, with reperfusion initiated after 2 h of ischemia. Neurological deficit was evaluated at 24 h after reperfusion, and scores were recorded for each group.

#### 2.3.4. Detection of Neurobehavioral Function Using Modified Neurological Severity Scores (mNSS)

The modified neurological severity score (mNSS), which covers motor, sensory, balance, and reflex functions, was used to evaluate the neurological status of rats in each group at 3 days after reperfusion. The score ranges from 0 to 18, with higher scores indicating more severe neurological impairment. All evaluations were independently performed by two researchers blinded to the experimental grouping to ensure result objectivity.

#### 2.3.5. Detection of Cerebral Infarct Volume by TTC Staining

At 24 h after reperfusion, rats were anesthetized with 1% pentobarbital sodium (50 mg/kg) via intraperitoneal injection, then sacrificed, and their brains were harvested. The brains were rinsed with pre-cooled PBS to remove blood, frozen at −20 °C for 30 min, and cut into consecutive coronal sections (2 mm thick) starting from the olfactory bulb using a brain mold. Sections were incubated with 2% TTC solution at 37 °C in the dark for 30 min (flipping every 10 min), fixed with 4% paraformaldehyde for 24 h, and photographed. The percentage of cerebral infarct volume was calculated using Image-Pro Plus 8.0 software.

#### 2.3.6. Observation of Neuronal Damage by Nissl Staining

Paraffin Sections of ischemic cerebral cortex were prepared, dewaxed with xylene, and dehydrated through gradient alcohols. Sections were stained with 1% toluidine blue solution, incubated in a 60 °C incubator for 40 min, rinsed with distilled water 3 times (3 min each), and differentiated with 95% alcohol. After gradient alcohol dehydration and xylene clearing, sections were mounted, observed under an inverted microscope (Zeiss, Oberkochen, Germany) at ×200 and ×400 magnification, and photographed. Normal neurons contain abundant Nissl bodies, while damaged neurons exhibit reduced, disintegrated, or absent Nissl bodies.

#### 2.3.7. Observation of Cerebral Cortex Histomorphology by HE Staining

Rats were anesthetized and sacrificed, and brains were harvested, fixed with 4% neutral formaldehyde for 48 h, dehydrated with gradient ethanol, cleared with xylene, embedded in paraffin after wax infiltration, and sectioned into 5 μm slices using a microtome (Leica, Vizsla, Germany). After dewaxing and rehydration, sections were stained with HE, dehydrated with gradient ethanol, cleared with xylene, and mounted. Pathological images of the ischemic penumbra were observed and collected under a light microscope (Leica, Vizsla, Germany).

#### 2.3.8. Observation of Mitochondrial Morphology in Cerebral Cortex by TEM

Fresh tissues were harvested, fixed with 2.5% glutaraldehyde, rinsed with PBS, dehydrated with a gradient of ethanol, embedded in epoxy resin, and polymerized in an oven. Ultra-thin sections (60–80 nm) were prepared using an ultramicrotome (Leica, Vizsla, Germany), stained with uranyl acetate for 30 min, dried overnight at room temperature, and observed and imaged using a transmission electron microscope (Jeol, Tokyo, Japan).

#### 2.3.9. Detection of ATP, ROS Levels, mPTP, and MMP by Kits

ATP and ROS levels were detected using commercial kits. ATP Detection Method: Weigh brain tissue and add pre-cooled double-distilled water to create a 1:9 mixture. Homogenize in an ice bath, then boil for 10 min before centrifuging at 3500 rpm for 10 min to obtain the supernatant. Prepare working solutions A and B according to the kit instructions. After grouping samples, incubate them in a 37 °C water bath. Following color development and termination reactions, measure absorbance at 636 nm and calculate content using the formula (expressed as μmol/g protein). ROS Experiment Method: Dissect brain tissue, digest with enzymes, filter through a 300-mesh nylon mesh, and centrifuge at 500× *g* for 10 min to obtain a single-cell suspension. Resuspend cells in 10 μM DCFH-DA working solution. Incubate at 37 °C for 30 min, then wash. Measure fluorescence intensity at 488 nm excitation and 525 nm emission wavelengths, with results expressed as relative fluorescence values. Mitochondria were isolated using a mitochondrial extraction kit, and their concentration was measured by the BCA method. The degree of the mPTP opening was determined using the mitochondrial swelling method. After extracting mitochondria, the protein concentration was adjusted to 10 mg/mL by the BCA method. A 20 μL of mitochondrial suspension was added to a 96-well plate, followed by 170 μL of buffer (reagent A), and the absorbance at 0 min was recorded. Then, 10 μL of swelling solution (reagent B) was added, and the absorbance change within 10 min was dynamically recorded using a multi-functional microplate reader at room temperature. The actual value was the difference between the absorbance at 0 min and 10 min (higher values indicate more obvious mitochondrial swelling). The level of MMP was detected by a multi-functional microplate (Thermo, Waltham, USA) reader according to the instructions.

#### 2.3.10. Immunofluorescence Detection of ROS, Bcl-2, and Bax in Rat Cerebral Cortex

After dewaxing sections to water, for ROS detection, sections were rewarmed, incubated with 10 μM DCFH-DA at 37 °C in the dark for 20 min, and rinsed with PBS 3 times. For Bcl-2/Bax staining, sections were repaired with 0.01 M sodium citrate in a microwave oven at medium power for 10 min, rinsed with PBS 3 times, blocked with 5% BSA for 30 min, and incubated with a mixed solution of Bcl-2 (1:100) and Bax (1:100) antibodies at 4 °C overnight. After rinsing with TBST 3 times (10 min each), sections were incubated with a mixed solution of Alexa Fluor 488 goat anti-rabbit (1:500) and 594 goat anti-mouse (1:500) secondary antibodies in the dark at room temperature for 1 h, stained with DAPI (1:1000) for 5 min, rinsed with PBS 2 times, mounted with anti-fluorescence quenching medium, and dried in the dark. Images were observed and captured under a fluorescence microscope (olympus, Tokyo, Japan): ROS and Bcl-2 showed red fluorescence, BAX showed green fluorescence, and cell nuclei showed blue fluorescence. Cells were analyzed using Image-Pro Plus 8.0 software.

#### 2.3.11. Detection of Parkin, PINK1, Bcl-2, Bax, and p62 Protein Expression by Western Blot

At 24 h after reperfusion, rats were anesthetized, and brains were harvested. Ischemic penumbra tissues were dissected on ice, washed, aliquoted into cryovials, and stored at −80 °C. Tissues were homogenized in 1 mL lysis buffer, incubated on ice for 15 min, centrifuged at 13,000 rpm for 10 min, and supernatants were collected. Protein concentration was measured by the BCA method. SDS-PAGE electrophoresis was performed, followed by membrane transfer and blocking with skimmed milk. Primary antibodies (β-actin, Parkin, Bcl-2, Bax, and p62, 1:1500) were added and incubated overnight at 4 °C. Membranes were rinsed with TBST 3 times, incubated with goat anti-rabbit secondary antibody (1:2500) at room temperature for 2 h, rinsed again, and developed with ECL. Band gray values were measured using Image-Pro Plus 8.0 software, and relative expression levels were expressed as the ratio of target protein to β-actin.

#### 2.3.12. Molecular Docking of AS-IV with PINK1 and Parkin

PINK1, Parkin were performed using AutoDockTools 1.5.6 and AutoDock Vina 4.2. The ligand structure was obtained from the PubChem database, optimized for 3D structure by energy minimization using ChemOffice14.0, and converted to PDBQT format with hydrogenation using AutoDockTools. Meanwhile, target crystal structures from RCSB PDB were processed using PyMOL 2.5.2 to remove solvents, add hydrogen, and define binding pockets. Specific site docking grids (PDBQT format) generated and exported by AutoDockTools were used for semi-flexible docking with AutoDock Vina 1.2.3, and binding affinity was quantified by free energy calculation. The final binding modes and molecular interactions were visualized using PyMOL.

### 2.4. Statistical Analysis

Experimental data were expressed as mean ± standard deviation (mean ± SD) and analyzed using SPSS 24.0 software. Comparisons between two groups were performed using the *t*-test, and comparisons among multiple groups were performed using one-way analysis of variance (ANOVA). GraphPad Prism 8.0.2 software was used for statistical analysis. *p* < 0.05 was considered statistically significant.

## 3. Results

### 3.1. AS-IV Improves Cerebral Infarct Volume and Neurological Function

After MCAO/R Injury, the study design is illustrated in Figure 1A. Rats were adaptively fed for 1 week, fasted before surgery, subjected to modeling (2 h of MCAO), administered corresponding treatments (AS-IV) at the start of reperfusion, followed by neurological function assessment and sample collection at 24 h. Figure 1B shows the changes in cerebral blood flow before, 2 h after, and after removal of the plug by infrared thermography to verify the influence of modeling on cerebral blood flow. Figure 1C quantifies MCAO/R and midbrain blood flow changes by dynamic curve. Results in Figure 1D demonstrate that compared to the Sham group. The 24-h mNSS score in the MCAO/R group was significantly increased (*p* < 0.01), whereas the 24-h mNSS scores in the AS-IV and NIM groups were significantly decreased compared to the MCAO/R group (*p* < 0.01). In Figure 1E, the Zea Longa score in the MCAO/R group was significantly higher than that in the Sham group (*p* < 0.01). In contrast, the Zea Longa scores in the AS-IV and NIM groups were significantly lower than those in the MCAO/R group (*p* < 0.01). Figure 1F shows the TTC staining results of brain tissues in each group, intuitively displaying differences among treatment groups. Figure 1G indicates that the percentage of cerebral infarct volume in the MCAO/R group was significantly higher than that in the Sham group (*p* < 0.01). Compared to the MCAO/R group, the AS-IV and NIM groups showed a significant reduction in infarct volume percentage (*p* < 0.01). To sum up, the AS-IV score of the Zea Longa decreased by 38.6% and the mNSS score decreased by 42.3%. The proportion of cerebral infarction volume decreased from 35.2% ± 3.1% to 14.5% ± 2.0%, decreasing by 58.8%, which was similar to that of the NIM group (*p* < 0.01). These results suggest that AS-IV exerts a protective effect against CIRI, improving neurological deficits and reducing infarct volume.

### 3.2. AS-IV Alleviates Neuronal and Cerebral Tissue Pathological AS-IV Damage After MCAO/R Injury

Nissl staining results (Figure 2A) showed that neurons in the Sham group had normal morphology with abundant and clear Nissl bodies; the MCAO/R group exhibited severe neuronal damage, characterized by reduced and disorganized Nissl bodies; the AS-IV and NIM groups showed alleviated neuronal damage, with improved morphology and quantity of Nissl bodies compared to the MCAO/R group. HE staining results (Figure 2B) revealed intact and regularly arranged cerebral cortex cells in the Sham group; the MCAO/R group presented pathological changes, such as enlarged intercellular spaces and disorganized structures. The AS-IV and NIM groups showed mitigated pathological damage, with relatively improved cell arrangement and structure compared to the MCAO/R group. These findings indicate that AS-IV has potential neuroprotective effects against MCAO/R-induced injury.

### 3.3. AS-IV Improves Mitochondrial Function and Cellular Ultrastructure After MCAO/R Injury

To clarify the effect of AS-IV on mitochondrial dysfunction induced by MCAO/R injury, transmission electron microscopy (TEM) was used to observe mitochondrial ultrastructure in the ischemic penumbra. TEM results (Figure 3A) showed normal cellular ultrastructure in the Sham group; in contrast, the MCAO/R group exhibited mitochondrial swelling and cristae disappearance, while the AS-IV and NIM groups showed reduced mitochondrial damage, with alleviated swelling and improved cristae integrity compared to the MCAO/R group. For ROS (Figure 3B,F) and mPTP (Figure 3D,H) detection, ROS levels and mPTP opening degree in the MCAO/R group were significantly higher than those in the Sham group (*p* < 0.01). Compared to the MCAO/R group, the AS-IV, NIM (for Figure 3B,D), and FCCP (for Figure 3F,H) groups showed significant reductions, with AS-IV reducing ROS by 47.5% and mPTP opening degree by 52.1% (*p* < 0.01). In the ATP (Figure 3C,G) and MMP (Figure 3E,I) assays, the MCAO/R group exhibited significantly lower ATP levels and MMP fluorescence intensity than the Sham group (*p* < 0.01). The AS-IV, NIM (for Figure 3C,E), and FCCP (for Figure 3G,I) groups showed significant increases, with AS-IV elevating ATP by 2.1-fold and MMP fluorescence intensity by 65.3% (*p* < 0.01).

### 3.4. AS-IV Alleviates MCAO/R Injury by Regulating ROS, Bcl-2, and BAX

Relevant indicators were detected to investigate the mechanism of AS-IV in CIRI (Figure 4). Compared to the Sham group, the MCAO/R group showed a significant increase in the proportion of ROS-positive cells (Figure 4A,B) (*p* < 0.01), a significant decrease in anti-apoptotic protein Bcl-2-positive cells (*p* < 0.01), and a significant increase in pro-apoptotic protein BAX-positive cells (Figure 4C) (*p* < 0.01), indicating activated oxidative stress and apoptotic pathways. After AS-IV intervention, the AS-IV and NIM groups exhibited a significant reduction in ROS-positive cells (*p* < 0.01), increased Bcl-2-positive cells (*p* < 0.01), and decreased BAX-positive cells (*p* < 0.01). These results suggest that AS-IV alleviates CIRI and exerts neuroprotective effects by inhibiting oxidative stress, regulating apoptotic proteins (upregulating Bcl-2 and downregulating BAX), with efficacy comparable to NIM.

### 3.5. AS-IV Regulates PINK1/Parkin and Apoptotic Proteins to Alleviate MCAO/R Injury

To further explore the mechanism, we detected mitophagy-related proteins (PINK1, Parkin, and p62) and apoptosis-related proteins (Bcl-2, BAX) (Figure 5). Across panels A–E (without FCCP) and F–J (with FCCP), the Sham group-maintained baseline protein expression, while the MCAO/R group showed disrupted mitophagy, with significantly decreased PINK1/Parkin, sharply increased p62 (*p* < 0.01), and activated apoptosis, characterized by elevated BAX and reduced Bcl-2 (*p* < 0.01). Notably, AS-IV exerted a robust therapeutic effect: compared to the MCAO/R group, AS-IV significantly restored PINK1 (2.3-fold) and Parkin (1.9-fold), reduced p62 (63.2% downregulation), lowered BAX (58.7% reduction), and elevated Bcl-2 (1.8-fold increase), with the Bcl-2/BAX ratio rising from 0.32 ± 0.04 to 1.15 ± 0.09 (*p* < 0.01) in both settings. These changes, comparable to NIM (A–E) and FCCP (F–J) effects, demonstrate AS-IV reverses MCAO/R-induced mitophagy and apoptosis disruptions, exerting neuroprotective effects against CIRI.

### 3.6. AS-IV Docking to PINK1/Parkin and Binding Energy Analysis

Molecular docking was performed to predict the potential binding modes between AS-IV and PINK1/Parkin proteins, providing structural biological clues for their activity regulation (Figure 6) to explore the regulatory mechanism of AS-IV. The results showed that AS-IV stably bound to specific active sites of PINK1 (Figure 6A, e.g., key residue regions such as Arg-30 and Ala-31) and Parkin (Figure 6B, e.g., Arg-82 and Lys-315), with binding energies of −10.14 kcal/mol and −9.36 kcal/mol, respectively. These precise binding sites and energy data support the interaction potential between AS-IV and PINK1/Parkin, laying a docking-based foundation for further investigating its potential regulatory role in PINK1/Parkin-mediated mitochondrial function.

## 4. Discussion

The high disability and mortality rates of ischemic stroke stem from its intricate pathological network, where mitochondrial dysfunction serves as the core hub linking oxidative stress, apoptosis, and neuronal death [27]. This study revealed that AS-IV synchronously ameliorates mitochondrial dysfunction and apoptosis imbalance by targeting and regulating the ROS-PINK1/Parkin pathway, providing a multi-dimensional mechanistic explanation for its neuroprotective effects. In recent years, although neuroprotective strategies targeting pathological links such as mitochondrial protection and inflammation regulation have made certain progress [27,28,29,30], a systematic intervention scheme has not yet been formed. By constructing a rat MCAO model, this study systematically evaluated the protective effects of AS-IV on ischemic brain injury. The results showed that AS-IV significantly improved neurological deficits, as evidenced by the decrease in Zea Longa scores and modified neurological severity scores (mNSS), reduced cerebral infarction volume, and alleviated neuronal injury and mitochondrial structure destruction. Meanwhile, it regulated the expression of proteins related to oxidative stress (decreased ROS levels) and apoptosis (increased Bcl-2/Bax ratio). This finding not only echoes the neuroprotective effects of AS-IV reported in previous studies but also reveals its improvement effects on ischemic stroke through quantitative indicators, providing more detailed evidence for clinical intervention.

After CIRI, mitochondrial respiratory chain dysfunction induces massive ROS accumulation [31]. As a dual-function signaling molecule, ROS directly damages proteins and RNA [32,33] and regulates mitophagy and apoptosis via molecular interactions [34,35]. In the MCAO/R group, suppressed PINK1/Parkin causes excessive apoptosis (elevated BAX, reduced Bcl-2), while AS-IV upregulates PINK1/Parkin, reversing these changes to coordinate mitophagy and apoptosis. By clearing excess ROS, AS-IV relieves PINK1/Parkin inhibition, restoring mitophagy (maintaining ATP and MMP) and inhibiting apoptosis via BAX/Bcl-2 regulation, achieving “anti-oxidation-promoting autophagy-anti-apoptosis” synergistic protection.

The core challenge in the current treatment of CIRI lies in how to balance neuroprotection and injury repair. Mitophagy has become a research hotspot due to its dual functions of clearing damaged mitochondria and maintaining energy metabolism. Previous studies have confirmed that PINK1/Parkin deficiency aggravates cerebral ischemic injury, while exogenous activation of this pathway can reduce the infarction volume [36,37,38]. However, there are controversies regarding the regulation of PINK1/Parkin by ROS. Some studies believe that ROS is a necessary signal for PINK1 activation [39,40], while others point out that excessive ROS will disrupt its function [41,42]. This study clearly shows that the ROS burst induced by ischemia–reperfusion turns into an inhibitory signal, and AS-IV precisely clears excessive ROS, which not only retains the physiological regulation of PINK1/Parkin by ROS but also avoids its toxic effects. This finding provides experimental evidence for analyzing the regulation of the ROS-PINK1/Parkin pathway.

As the core active component of traditional Chinese medicine, Astragalus membranaceus, the research on the neuroprotective effects of AS-IV has gradually become a hotspot in recent years [43,44,45]. Existing studies have confirmed that AS-IV can alleviate inflammatory responses by inhibiting the NF-κB pathway [46] or promote the survival of nerve cells by activating the Akt/mTOR pathway [47]. However, these mechanisms mostly focus on downstream effects and lack systematic regulation of cell homeostasis maintenance. This study directly associates its neuroprotective effects with the regulation of PINK1/Parkin-mediated mitochondrial function, breaking through the previous cognitive limitations of its action mechanisms. Compared to single antioxidant or anti—inflammatory strategies, AS-IV, through regulating the ROS-PINK1/Parkin pathway, achieves an upgrade from passive removal of damage to active maintenance of mitochondrial homeostasis, providing a more in-depth theoretical support for the clinical application of AS-IV. Meanwhile, the results of this study also provide a new perspective for the application and transformation of active components of traditional Chinese medicines in the field of modern neuroscience.

## 5. Conclusions

In summary, this study confirms that AS-IV exerts multi-target protective effects against ischemic stroke by regulating the ROS-PINK1/Parkin pathway to improve mitochondrial function and inhibit neuronal apoptosis. It provides experimental evidence for the clinical application of AS-IV. Future studies may further explore the synergistic effects of AS-IV with other neuroprotective strategies, optimize the administration regimen, and conduct preclinical safety evaluations to promote its translation from basic research to clinical treatment.

## 6. Limitations

This study has several limitations. Mechanistically, it did not verify via gene knockout if PINK1/Parkin are essential targets for AS-IV, leaving other pathways’ synergistic effects unruled out. For clinical translation, there are pathological differences between the animal model and human ischemic stroke; only a single AS-IV dose (20 mg/kg) was used, requiring further exploration of its dose-effect relationship and long-term safety. Additionally, without cell-specific analysis, it is unclear if AS-IV regulates the ROS-PINK1/Parkin pathway differently across cell types in the ischemic penumbra. Moreover, the study used only male rats, potentially limiting generalizability due to gender differences. It also lacks long-term follow-up data on the persistence of neuroprotective and reparative effects over time and has not investigated the influence of non-neuronal cells other than astrocytes (e.g., microglia) on AS-IV’s regulatory effects.

## Figures and Tables

**Figure 1 cimb-47-00597-f001:**
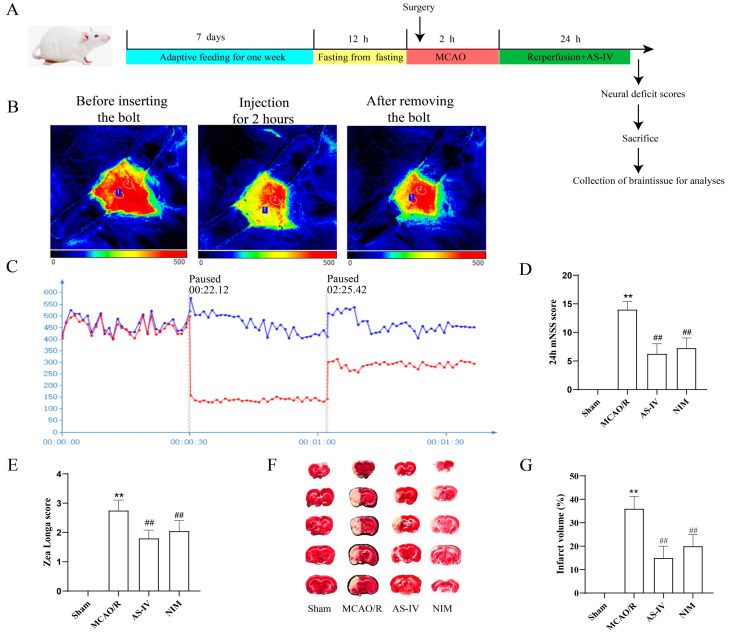
(**A**) Experimental design and outcome evaluations of AS-IV in a rat model of ischemic stroke. (**B**) Infrared thermography was used to monitor cerebral blood flow. (**C**) Brain blood flow curve, Blue represents the healthy side and red represents the affected side. (**D**) 24-h modified neurological severity score (mNSS) results. (**E**) Zea Longa score results. (**F**) Representative TTC-stained brain sections showing infarcted (pale) and non-infarcted (red) areas. (**G**) Quantitative analysis of cerebral infarct volume. ** *p* < 0.01 vs. Sham; ^##^
*p* < 0.01 vs. MCAO/R.

**Figure 2 cimb-47-00597-f002:**
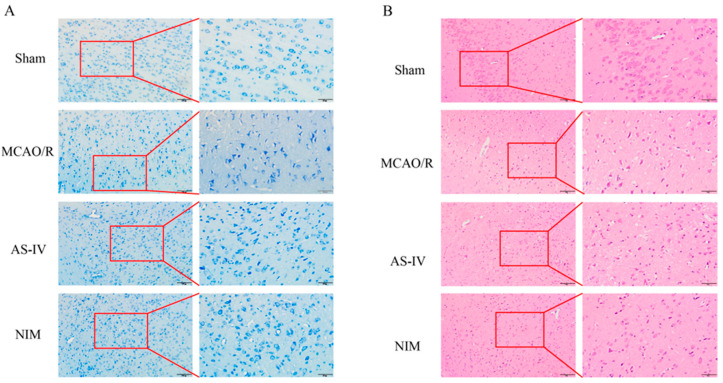
Histopathological changes in the brain tissues of rats in different groups. (**A**) Nissl staining results (200× and 400× magnification). The red-framed areas are magnified to display detailed cellular structures. (**B**) HE staining results (200× and 400× magnification). The red-framed regions are enlarged to present specific tissue and cellular alterations.

**Figure 3 cimb-47-00597-f003:**
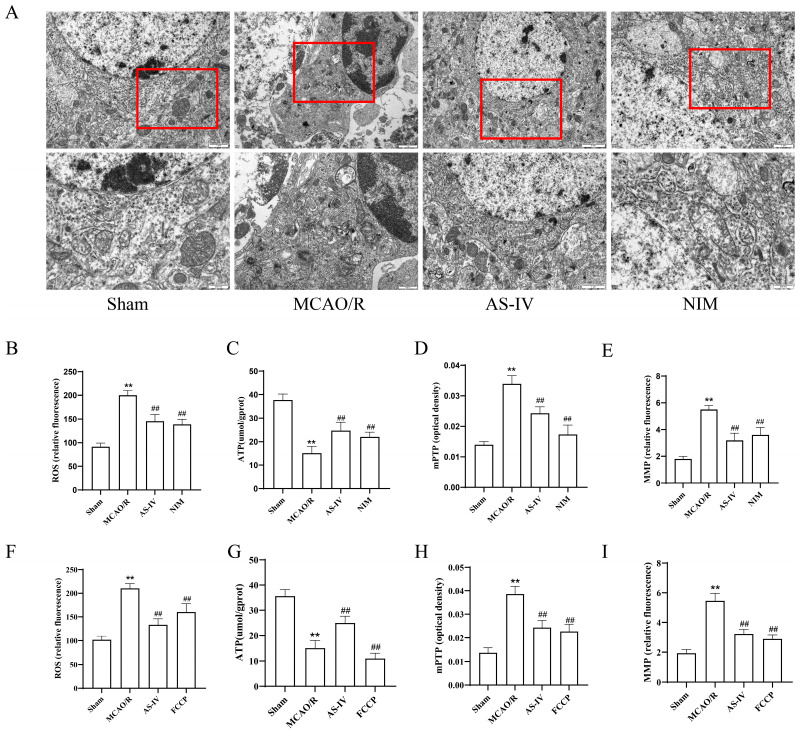
(**A**) TEM images showing mitochondrial morphology in rat cerebral cortex across groups.The red box is used to highlight specific mitochondrial regions that we focus on observing and analyzing. (**B**–**E**) Quantitative analyses of key mitochondrial/oxidative stress indicators. (**B**,**F**) ROS levels. (**C**,**G**) ATP content. (**D**,**H**) mPTP opening degree. (**E**,**I**) MMP Data. ** *p* < 0.01 vs. Sham; ^##^
*p* < 0.01 vs. MCAO/R.

**Figure 4 cimb-47-00597-f004:**
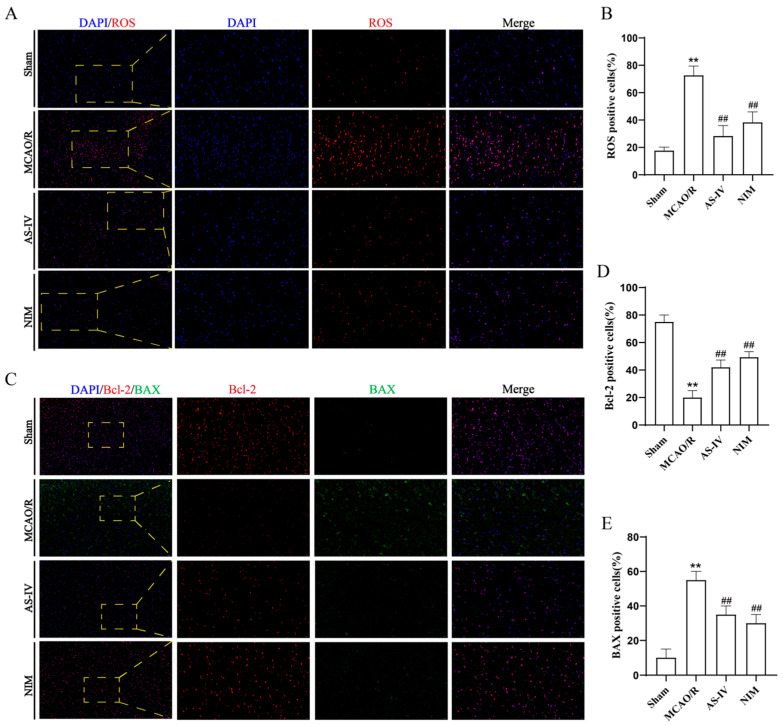
(**A**) Representative fluorescence images of ROS detection (red fluorescence) in brain tissues (×200 and ×400 magnification, Blue fluorescence represents DAPI). Yellow dashed boxes in the 200× images indicate regions magnified to ×400 to show detailed ROS-positive cell distribution. (**B**) Quantitative analysis of ROS-positive cell percentages. (**C**) Immunofluorescence staining results (×200 and ×400 magnification). Red fluorescence represents BCL-2, green fluorescence represents BAX and Blue fluorescence represents DAPI in brain tissues. Yellow dashed boxes in the ×200 images mark areas enlarged to 400× for clear visualization of BCL-2/BAX expression patterns. (**D**,**E**) Statistical results of BCL-2-positive cell percentages and BAX-positive cell percentages, reflecting the regulation of apoptosis-related proteins and oxidative stress by different treatments.** *p* < 0.01 vs. Sham; ^##^
*p* < 0.01 vs. MCAO/R.

**Figure 5 cimb-47-00597-f005:**
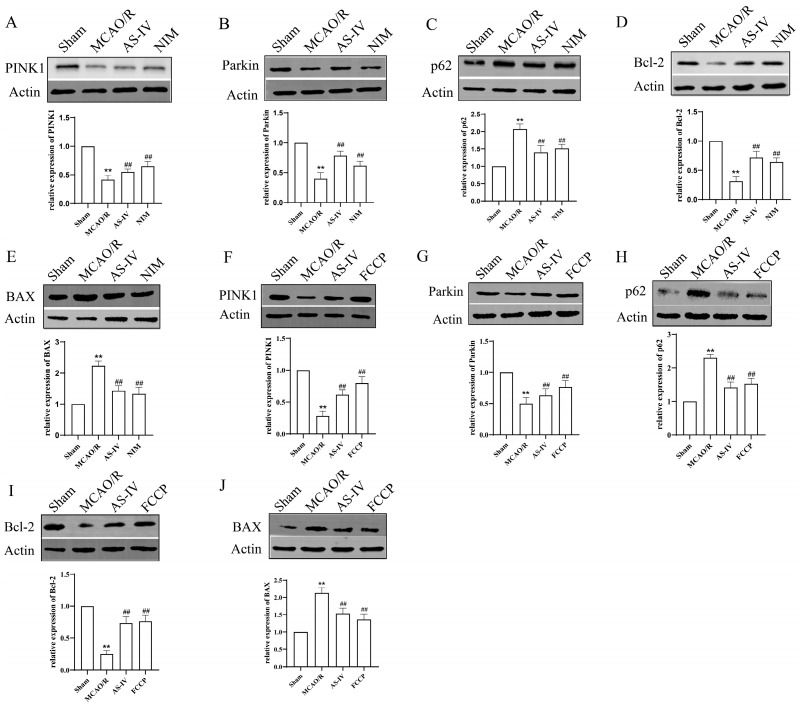
Western blot analysis of protein expression related to mitophagy and apoptosis in brain tissues of different groups. (**A**–**J**) Representative Western blot bands and quantitative results of PINK1, Parkin, p62, Bcl-2, and BAX proteins. β-Actin was used as an internal reference. ** *p* < 0.01 vs. Sham; ^##^
*p* vs. MCAO/R.

**Figure 6 cimb-47-00597-f006:**
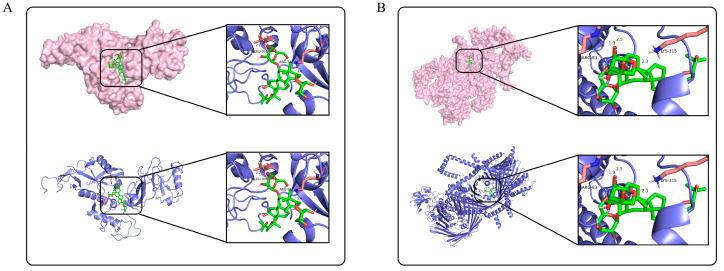
Molecular docking results of AS-IV with PINK1 and Parkin. (**A**) shows the molecular docking result of AS-IV with PINK1. (**B**) shows the molecular docking result of AS-IV with Parkin. The protein structures are displayed in surface model (pink) and ribbon model (blue), respectively, and the locally enlarged area presents the binding details between AS-IV (green) and key residues of the proteins.

## Data Availability

The data presented in this study are available on request from the corresponding author.

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
