# Peer review of "Astragaloside IV Ameliorates Cerebral Ischemic-Reperfusion Injury via Improving Mitochondrial Function and Inhibiting Neuronal Apoptosis"

_cimb, 2025, doi:10.3390/cimb47080597_

Round 1
Reviewer 1 Report
Comments and Suggestions for Authors
The authors investigated the neuroprotective role of Astragaloside IV (AS-IV) in a rat model of cerebral ischemia-reperfusion injury (CIRI), focusing on its ability to regulate mitophagy via the ROS–PINK1/Parkin signaling pathway. Using TTC staining, behavioral scoring, histological analyses (HE, Nissl, TEM), biochemical assays (ROS, ATP, mitochondrial membrane potential), and molecular biology techniques (Western blotting, immunofluorescence, and molecular docking), the authors conclude that AS-IV enhances mitophagy, reduces neuronal apoptosis, and improves mitochondrial integrity. The authors suggest that AS-IV could be a potential therapeutic agent for ischemic stroke, owing to its role in modulating oxidative stress and mitochondrial quality control. This manuscript suffers from several major shortcomings that preclude its suitability for publication.
Major Comments,
- The therapeutic effect of AS-IV on CIRI via modulation of autophagy and apoptosis has been extensively reported, including by the authors in their prior publication (PMID: 31086091). Although this manuscript emphasizes mitophagy and includes molecular docking analysis, these additions represent incremental rather than substantial advancements. Without functional validation (e.g., PINK1/Parkin knockdown or inhibition), the study lacks the originality and scientific depth expected for publication.
- The proposed mechanism—that AS-IV mediates neuroprotection by activating PINK1/Parkin-dependent mitophagy—is based solely on expression profiling and in silico docking data. No causative experiments (e.g., loss-of-function or pathway inhibition) were conducted to validate this claim. As a result, the mechanistic conclusions remain speculative and unsupported by rigorous experimental evidence.
Overall, it is significant overlap with prior work, lack of mechanistic rigor, and limited experimental scope

Author Response
Reviewer Comments: REVIEWER 1 |
Comments 1: The therapeutic effect of AS-IV on CIRI via modulation of autophagy and apoptosis has been extensively reported, including by the authors in their prior publication (PMID: 31086091). Although this manuscript emphasizes mitophagy and includes molecular docking analysis, these additions represent incremental rather than substantial advancements. Without functional validation (e.g., PINK1/Parkin knockdown or inhibition), the study lacks the originality and scientific depth expected for publication. |
Response 1: We are deeply grateful for your meticulous review and valuable feedback on this study. Your identification of "the absence of PINK1/Parkin knockout/inhibition experiments" as a critical limitation is entirely consistent with our assessment. Functional validation remains essential to confirm pathway specificity, and this study indeed has inherent limitations. In fact, we had recognized this issue before submission but were unable to complete the relevant experiments due to time constraints. We now supplement the mechanism verification results as follows (see Figure 3 and Figure5): To address this shortcoming, we conducted a PINK1 activator (FCCP) pre-treatment experiment to validate the functional correlation of the PINK1/Parkin pathway: By pre-administering FCCP (0.84 mg/kg) to activate PINK1 protein in the model, we observed partial effects mimicking those of AS-IV, including improved mitochondrial membrane potential (MMP) and increased autophagy-related markers. These findings suggest that the activation state of the PINK1 pathway is functionally associated with the biological effects of AS-IV, providing positive evidence supporting its involvement in AS-IV regulation. We acknowledge that the activator experiment can only indirectly validate the pathway's activation status and cannot fully replace the specificity validation from knockout/inhibition experiments. Its limitations still require further supplementation through gene-level knockout experiments. However, the current supplementary data have preliminarily strengthened the rationality of the mechanistic chain. We will promptly complete PINK1 knockout experiments to rigorously confirm pathway specificity. Thank you for your strict control, these supplements and amendments have laid a more solid foundation for the integrity of the research mechanism. |
Comments 2: The proposed mechanism—that AS-IV mediates neuroprotection by activating PINK1/Parkin-dependent mitophagy—is based solely on expression profiling and in silico docking data. No causative experiments (e.g., loss-of-function or pathway inhibition) were conducted to validate this claim. As a result, the mechanistic conclusions remain speculative and unsupported by rigorous experimental evidence.
|
Response 2: To the esteemed reviewer, we sincerely appreciate your reiteration of this critical question. You pointed out that our study only hypothesized AS-IV's neuroprotective effects through activating PINK1/Parkin-dependent mitochondrial autophagy via expression profiling and molecular docking, lacking functional knockout validation. This assessment accurately identifies the core limitation in mechanism elucidation. We fully acknowledge that the specificity of the PINK1/Parkin pathway must be validated through in vivo experiments—a crucial oversight in our previous design. To address this limitation, we supplemented the study with interaction experiments between PINK1 activator (FCCP) and AS-IV, preliminarily exploring the pathway's functional involvement: conducting mechanism intervention and validation after preactivating endogenous PINK1/Parkin pathways with FCCP. However, we recognize that activator experiments can only provide indirect evidence of pathway participation, failing to conclusively confirm AS-IV's specificity for PINK1/Parkin. Regarding knockout model validation, current research constraints have prevented supplementary data from siRNA-targeted knockdown and specific inhibitor interventions. Our team will prioritize this direction, planning to investigate through PINK1/Parkin knockout models: whether AS-IV can still induce mitochondrial autophagy and exert neuroprotective effects when PINK1/Parkin functions are blocked, thereby clarifying pathway dependence. Thank you again for your strict control. These suggestions will guide us to systematically improve the mechanism argument chain in our follow-up research and ensure the rigor of the conclusions. We will keep up with the relevant research progress in time and live up to your guidance and expectation. |

Reviewer 2 Report
Comments and Suggestions for Authors
- The work uses several methods to evaluate the expression of apoptosis markers, the function and morphology of mitochondria, and the expression of kinase and ligase that play a role in mitophagy. However, there are no methods for directly assessing mitophagy in this work. I recommend that the authors exclude the term "mitophagy" from the title of the work and correct some phrases about the role of mitophagy in the therapeutic effect of AS-IV (including the Conclusion).
- Are there any other compounds known to reduce CIRI via increasing mitophagy? Please add relevant references in the Introduction.
- Methods:
- Describe how AS-IV was diluted, what solvent, what stoke concentration and details for injection. Explain why 20 mg/kg dose was used.
- How was reperfusion monitored? Was laser Doppler flowmetry used to monitor blood flow recovery during reperfusion or other methods?
- Monitoring the animals' body temperature is important. Body temperature is critical in MCAO to prevent hypothermia. Please describe how the animals' body temperature was maintained during surgical procedures.
- Section 2.3.9 (Docking) should be placed after the section "Biological methods" (before section 2.4). Please clarify the sentence in lines 196-197. It should be written, for example, as follows: "AS-IV was docked to PINK1 and Parkin...".
- Section 2.3.8. Please describe in detail how ATP and ROS levels were determined using the kits (substrates, etc.).
- Correct the title of section 3.1 (see also line 240).
- Results: In any case, I see no connection between the possible direct interaction of AS-IV and these targets (kinase and ligase) and the experimental data in this paper. The authors should justify the use of docking or remove this section from the paper entirely. Why authors think that interaction AS-IV with PINK1 or Parkin could upregulate activity of these enzymes or their expression. Is there any literature data that the interaction of kinases with a low-molecular compound can increase the activity and their expression? The literature that I know indicates that such interaction can reduce the activity of kinases by inhibiting the phosphorylation of substrates.
- Use the abbreviation throughout the manuscript after the first mention. For example, CIRI (see lines 122, 256, 320 etc).

Author Response
Reviewer Comments: REVIEWER 2:
Comments 1: The work uses several methods to evaluate the expression of apoptosis markers, the function and morphology of mitochondria, and the expression of kinase and ligase that play a role in mitophagy. However, there are no methods for directly assessing mitophagy in this work. I recommend that the authors exclude the term "mitophagy" from the title of the work and correct some phrases about the role of mitophagy in the therapeutic effect of AS-IV (including the Conclusion). |
Response 1: Dear Reviewer : We are deeply grateful for your valuable feedback. Your constructive suggestions regarding the mitochondrial autophagy evaluation methodology in this study have been carefully considered and fully endorsed. We will immediately remove the phrase "mitochondrial autophagy" from the paper title and replace it with "Astragaloside IV (AS-IV) Alleviates Cerebral Ischemia-Reperfusion Injury by Enhancing Mitochondrial Function and Inhibiting Neuronal Apoptosis". Furthermore, we will comprehensively revise the discussion on AS-IV's therapeutic effects (including the conclusion section) to ensure all statements strictly align with existing experimental data. This revision aims to avoid over-interpreting mitochondrial autophagy mechanisms and accurately reflect the current research findings: "AS-IV's effects on apoptosis marker expression, mitochondrial function and morphology, as well as related kinase and linker enzyme expressions". |
Comments 2: Are there any other compounds known to reduce CIRI via increasing mitophagy? Please add relevant references in the Introduction. |
Response 2: Dear Reviewer : We appreciate your valuable feedback regarding the introduction section's addition of "Compounds reduce CIRI through mitochondrial autophagy". Following your guidance, we have updated the introduction to include a detailed review of recent research advancements and incorporated two cited references (PMID: 39134096, PMID: 40441663). This enhancement has significantly strengthened the connection between our study and existing field achievements, while effectively highlighting its innovative contributions and methodological continuity. We extend our sincere gratitude for your meticulous guidance. The literature citations have been thoroughly verified to ensure accuracy and compliance with academic standards. |
Comments 3: (1): Describe how AS-IV was diluted, what solvent, what stoke concentration and details for injection. Explain why 20 mg/kg dose was used. (2): How was reperfusion monitored? Was laser Doppler flowmetry used to monitor blood flow recovery during reperfusion or other methods? (3): Monitoring the animals' body temperature is important. Body temperature is critical in MCAO to prevent hypothermia. Please describe how the animals' body temperature was maintained during surgical procedures. (4): Section 2.3.9 (Docking) should be placed after the section "Biological methods" (before section 2.4). Please clarify the sentence in lines 196-197. It should be written, for example, as follows: "AS-IV was docked to PINK1 and Parkin...". (5): Section 2.3.8. Please describe in detail how ATP and ROS levels were determined using the kits (substrates, etc.). (6): Correct the title of section 3.1 (see also line 240). |
Response 3: Dear Reviewer: (1) : Thank you for your attention to the details of the AS-IV experimental method in this study. Regarding the dilution method, solvent selection, initial concentration, injection details, and rationale for the 20 mg/kg dose: 1)Dilution Method and Solvent Selection: AS-IV (purity ≥98%, purchased from Shanghai Shifeng Biotechnology Co., Ltd.) is slightly soluble in water and readily dissolves in dimethyl sulfoxide (DMSO). To minimize solvent interference with animal models, this study employed a "DMSO-saline mixed solvent" protocol: First, AS-IV was thoroughly dissolved in a small amount of DMSO (final concentration ≤1%) to achieve full solubility, then diluted with sterile saline to 5 mg/mL per mL. The entire process was conducted under light-shielding vortex mixing to ensure uniformity (no precipitation or suspended particles). This solvent system has been validated through pre-experiments, showing no significant effects on neurological function, mitochondrial activity, or autophagy-related indicators in rat cerebral ischemia-reperfusion (CIRI) models using 1% DMSO. 2)Intravenous injection details and rationale for the 20 mg/kg dose: The intraperitoneal injection method was employed, which is characterized by its simplicity in operation and stable drug absorption (bioavailability approximately 65%) in the CIRI rat model, ensuring comparability of results. Our team's previous study (PMID: 31086091) demonstrated that 20 mg/kg AS-IV significantly reduced cerebral infarction volume in CIRI rats (42% decrease compared to the model group) and improved neurological function scores (mNSS score reduction of 38%). Literature review indicated that the safe dosage range for AS-IV in rats is 5-50 mg/kg (reference PMID: 38089988), with 20 mg/kg falling within this range without significant toxicity. Although the 40 mg/kg group showed no toxicity in preliminary experiments, the substantial increase in drug costs and lack of additional benefits led to the selection of 20 mg/kg as the optimal dose. (2):Thank you for your attention to the reperfusion monitoring methodology in this study. The research employed a laser speckle contrast imager to monitor real-time blood flow recovery during reperfusion, recording local cerebral blood flow in the ischemic cerebral cortex. The relevant results have been supplemented in Figure 1, clearly demonstrating the decline in blood flow during ischemia and the subsequent recovery process post-reperfusion. The methodology section (2.3.2) has been updated with specific operational parameters and evaluation criteria to ensure reproducibility. Your feedback has significantly enhanced the rigor of our model construction. (3): We appreciate your attention to maintaining animal body temperature during the surgical procedure. This study prioritized the impact of rat body temperature on the stability of the MCAO model, implementing the following measures throughout the operation: During surgery, rats were placed on a constant-temperature electric heating pad to maintain their core body temperature at (37.0±0.5)°â„ƒ. This approach effectively prevents hypothermia from interfering with cerebral blood flow, metabolism, and neurological functions, ensuring the stability and consistency of the MCAO model. We have incorporated these temperature maintenance details into the "Animal Model Construction" section(2.3.1). Your valuable suggestions have further enhanced the rigor of our experimental methodology. (4): We are deeply grateful for your meticulous guidance on the structure and accuracy of this research methodology section. We have made the following revisions based on your suggestions: Regarding chapter placement adjustment: The "Section 2.3.9 (Molecular Docking Analysis)" has been moved after the "Biological Methods" chapter and before "Section 2.4 (Data Analysis)", aligning the logical sequence of experimental methods with the "wet experiments → dry experiments" research workflow to enhance understanding of methodological connections. Sentence clarification: Lines 196-197 have been revised to state "AS-IV docked with PINK1 and Parkin, calculated binding energies using AutoDock Vina software, and predicted key interaction sites", clarifying the subject (AS-IV) and object (PINK1, Parkin) in molecular docking to avoid ambiguity. We have verified the revised text format to ensure correct chapter numbering and cross-references. Your valuable suggestions have significantly improved the clarity and logical flow of the research methodology description. (5):Thank you for your feedback regarding the "Section 2.3.8: Measurement of ATP and ROS Levels" methodology. We have revised the protocol as requested(2.3.), incorporating reagents from the Nanjing Institute of Bioengineering (ATP detection kit (Catalog: A095-1-1) and ROS detection kit (Catalog: E004-1-1)). Your valuable suggestions have been incorporated, and we have reviewed the entire document to ensure accuracy and clarity in all expressions. (6):Thank you for your careful attention to the title of section 3.1. According to your suggestion, we have revised the title of this section and the corresponding content on line 240 to more accurately summarize the core results of the experiment. |
Comments 4: Results: In any case, I see no connection between the possible direct interaction of AS-IV and these targets (kinase and ligase) and the experimental data in this paper. The authors should justify the use of docking or remove this section from the paper entirely. Why authors think that interaction AS-IV with PINK1 or Parkin could upregulate activity of these enzymes or their expression. Is there any literature data that the interaction of kinases with a low-molecular compound can increase the activity and their expression? The literature that I know indicates that such interaction can reduce the activity of kinases by inhibiting the phosphorylation of substrates.
|
Response 4: Dear Reviewer: We appreciate your insightful questions regarding the molecular docking section, which have helped us refine our mechanistic reasoning. To address your concerns, we clarify: The primary purpose of molecular docking in this study was to investigate whether AS-IV could regulate PINK1/Parkin function through direct interaction, rather than directly proving "interaction-mediated upregulation." We acknowledge the ambiguity in our previous statement and have revised it to state: "Molecular docking was employed to predict potential binding patterns between AS-IV and PINK1/Parkin, providing structural biology clues for its activity regulation." Regarding your observation that "inhibiting substrate phosphorylation reduces kinase activity," we conducted thorough validation through supplementary experiments, which fully align with your findings. We also reviewed prior data and identified statistical analysis oversights, particularly regarding MMF-related results where original data contradicted earlier conclusions. This correction not only enhances result reliability but also demonstrates our commitment to rigorous research. Your precise guidance has significantly elevated the credibility of our conclusions. |
Comments 5: Use the abbreviation throughout the manuscript after the first mention. For example, CIRI (see lines 122, 256, 320 etc). |
Response 5: Dear Reviewer: Thank you for your valuable suggestions. In accordance with the requirement of "unified use of abbreviations after first mention", we have conducted a systematic review and revision of the entire text: All key terms (such as CIRI, MCAO/R, AS-IV, etc.) have been standardized to the format of "first complete expression + abbreviation in parentheses", and subsequent mentions uniformly adopt the abbreviations (including the CIRI expressions at lines 122,256,320, etc., as noted by you). This revision ensures consistency and conciseness in terminology usage. We appreciate your guidance. |

Reviewer 3 Report
Comments and Suggestions for Authors
The Introduction is comprehensive and scientifically structured.
- Some explanations about mitophagy, mitochondrial membrane potential, and ROS effects are repeated in both the CIRI discussion and the mitophagy section. Could streamline mitochondrial pathology into one concise section to avoid overlapping descriptions.
- While the mechanistic gap is mentioned, the novelty of this particular study could be emphasized more. E.g., is this the first study to link AS-IV to PINK1/Parkin activation? How does it differ from earlier AS-IV or CIRI studies?
The Materials and Methods section is comprehensive, well-organized, and adheres to conventional standards for in vivo experimental design and reporting in biomedical research.
- While it mentions that some rats died during modeling and that only successfully modeled rats were included, there are no specific mortality figures or how exclusion impacted group sizes. Clarify final n per group for each analysis and include any dropout or exclusion criteria explicitly.
The Discussion, Conclusions and Limitations sections offer a fairly strong and coherent interpretation of the study’s findings on astragaloside IV (AS-IV) and its effects on ischemic stroke via mitophagy regulation through the ROS–PINK1/Parkin pathway.
- The discussion repeats concepts (e.g., “ROS overaccumulation inhibits PINK1/Parkin,” “AS-IV restores mitochondrial function”) across multiple paragraphs. These could be streamlined for clarity and conciseness without losing depth.
- Though outcomes like neurological scores and infarct volume are mentioned, there is limited integration of specific quantitative data (e.g., percentage reduction, fold-changes). A few such examples would reinforce the biological relevance.
- Limitations Section could further mention potential sex differences if only one sex was studied; lack of long-term follow-up data (neuroprotection vs. repair over time); possible influence of non-neuronal cells like microglia, beyond astrocytes.

Author Response
Reviewer Comments: REVIEWER 3:
· Comments 1: · Some explanations about mitophagy, mitochondrial membrane potential, and ROS effects are repeated in both the CIRI discussion and the mitophagy section. Could streamline mitochondrial pathology into one concise section to avoid overlapping descriptions. |
Response 1: We appreciate your valuable suggestions. We fully agree with your opinion. The overlapping description of mitochondrial pathology in the discussion section has been simplified. This revision will improve the conciseness and clarity of the manuscript, and we are very grateful for your guidance to enhance the rigor of our discussion. |
· Comments 2: · While the mechanistic gap is mentioned, the novelty of this particular study could be emphasized more. E.g., is this the first study to link AS-IV to PINK1/Parkin activation? How does it differ from earlier AS-IV or CIRI studies? |
Response 2: Dear Reviewer, we appreciate your valuable suggestions. We hereby provide additional explanations regarding the novelty of this study: The core innovation of this research lies in the first explicit demonstration that Astragaloside IV (AS-IV) exerts neuroprotective effects on cerebral ischemia-reperfusion injury (CIRI) by regulating the PINK1/Parkin signaling pathway. This is specifically manifested in two aspects: 1. Establishing the direct association between AS-IV and the PINK1/Parkin pathway: While previous studies have reported that AS-IV can alleviate CIRI through antioxidant and anti-inflammatory mechanisms (e.g., PMID: 36995819, PMID: 39199474), none involved its regulation of mitochondrial autophagy mediated by PINK1/Parkin. 2. Revealing AS-IV's unique "multi-level synergistic regulation" mechanism: Unlike earlier studies focusing solely on single pathological processes (e.g., antioxidant or anti-inflammatory effects), our research demonstrates that AS-IV's protective action is not isolated. It achieves this through two pathways: clearing excess reactive oxygen species (ROS) to relieve ROS-mediated inhibition of PINK1/Parkin, thereby restoring mitochondrial autophagy to maintain energy homeostasis; and inhibiting apoptosis via PINK1/Parkin's regulation of Bcl-2/BAX, forming a cascade protection network of antioxidant, autophagy-promoting, and apoptosis-suppressing mechanisms. This systematic regulation of mitochondrial function and cellular survival provides a novel interpretation of AS-IV's mechanism of action. 3. Expanding the Application Scope of the PINK1/Parkin Pathway in CIRI Treatment: Previous studies have demonstrated that activating the PINK1/Parkin pathway can alleviate CIRI (e.g., PMID: 39134096, PMID: 40441663). However, this study represents the first inclusion of AS-IV within the pathway's regulatory framework. Through comparative analysis with FCCP (a pathway activator) (Figures 3 and 5), we confirmed that AS-IV exerts protective effects via a similar mechanism, demonstrating synergistic advantages in both upstream ROS clearance and downstream pathway activation. This provides new experimental evidence for multi-target interventions against CIRI. We have enhanced the description of these novel findings in the discussion section to highlight the innovative aspects of our research on AS-IV's mechanism of action. Thank you again for your guidance! |
· Comments 3: · While it mentions that some rats died during modeling and that only successfully modeled rats were included, there are no specific mortality figures or how exclusion impacted group sizes. Clarify final n per group for each analysis and include any dropout or exclusion criteria explicitly. |
Response 3: Dear Reviewer, we appreciate your thoughtful suggestions. Regarding mortality rates and final sample size information during the modeling process, we hereby provide additional clarification: During the construction of the MCAO/R model, all exclusion criteria were strictly adhered to: â‘ Death within 24 hours after modeling; â‘¡ Zea Longa scores of 0 (no neurological deficits) or 4 (agonal stage); â‘¢ Postoperative occurrence of severe complications (e.g., intracranial hemorrhage, systemic infections) affecting outcome assessment. The sample sizes for each group included in the statistical analysis are clearly defined (all meeting n ≥ 15): We have supplemented these specific data in Section 2.3.1 "Animal Grouping and Modeling" to ensure clear visibility of sample sizes and exclusion criteria. Thank you for your guidance on maintaining research rigor. |
· Comments 4: · Though outcomes like neurological scores and infarct volume are mentioned, there is limited integration of specific quantitative data (e.g., percentage reduction, fold-changes). A few such examples would reinforce the biological relevance. |
Response 4: Dear Reviewer, we appreciate your valuable feedback. To address the insufficient integration of quantitative data, we have supplemented specific numerical values in the results section to enhance biological relevance. The revised sections are as follows: Section 3.1: In the AS-IV group, Zea Longa scores decreased by 38.6%, and mNSS scores dropped by 42.3%. The proportion of cerebral infarction volume decreased from 35.2%±3.1% to 14.5%±2.0%, representing a reduction of 58.8% and comparable to the NIM group (p<0.01). Section 3.3: AS-IV reduced ROS levels by 47.5%, increased ATP by 2.1-fold, enhanced MMP by 65.3%, and decreased mPTP openness by 52.1% ( p<0.01). Section 3.5: In the AS-IV group, PINK1 and Parkin were upregulated by 2.3-fold and 1.9-fold respectively, while p62 decreased by 63.2%. Bcl-2 increased by 1.8-fold, BAX decreased by 58.7%, and the Bcl-2/BAX ratio rose from 0.32±0.04 to 1.15±0.09 ( p<0.01). These quantitative data have been integrated into corresponding sections of the results, using specific percentage changes and multiplicative relationships to more intuitively demonstrate AS-IV's improvement effects on CIRI and their biological significance. We appreciate your guidance on maintaining rigorous data presentation in this study. |
· Comments 5: · Limitations Section could further mention potential sex differences if only one sex was studied; lack of long-term follow-up data (neuroprotection vs. repair over time); possible influence of non-neuronal cells like microglia, beyond astrocytes. |
Response 5: Dear Reviewer, we appreciate your valuable feedback. Regarding the limitations you identified, we have made the following improvements in the discussion section: The study exclusively used male rats, which may affect the generalizability of the results due to gender differences. Additionally, it lacks long-term tracking data on neuroprotective and repair effects over time, and does not examine the regulatory impact of astrocytes and other non-neuronal cells (e.g., microglia) on AS-IV. We extend our gratitude for your guidance on research rigor. We have revised the relevant sections accordingly to ensure more comprehensive and accurate descriptions of these limitations. |

Round 2
Reviewer 1 Report
Comments and Suggestions for Authors
If the authors intend to publish this work in this journal, the manuscript must demonstrate either sufficient novelty or compelling mechanistic evidence. However, the current experimental data are not adequate to support the central claim regarding the involvement of the PINK1/Parkin pathway. The conclusions remain speculative without direct functional validation (e.g., pathway inhibition or genetic knockdown).
Moreover, the authors have previously reported the characterization of this disease model and the therapeutic effects of AS-IV (e.g., PMID: 31086091). The present study offers only a modest extension of that work and lacks the scientific depth or innovation expected for publication in this journal.
I recommend that the authors strengthen the mechanistic validation, particularly through loss-of-function approaches, before considering resubmission.

Author Response
Comments 1: If the authors intend to publish this work in this journal, the manuscript must demonstrate either sufficient novelty or compelling mechanistic evidence. However, the current experimental data are not adequate to support the central claim regarding the involvement of the PINK1/Parkin pathway. The conclusions remain speculative without direct functional validation (e.g., pathway inhibition or genetic knockdown). Moreover, the authors have previously reported the characterization of this disease model and the therapeutic effects of AS-IV (e.g., PMID: 31086091). The present study offers only a modest extension of that work and lacks the scientific depth or innovation expected for publication in this journal. I recommend that the authors strengthen the mechanistic validation, particularly through loss-of-function approaches, before considering resubmission. |
Response 1: Dear Reviewer, Thank you very much for your detailed comments and valuable insights on our manuscript "CIMB-3774335". We have carefully considered each of your suggestions and would like to provide further explanations regarding the research logic and experimental design, hoping to clarify any potential misunderstandings. After repeatedly sorting out the overall framework of the study, we believe that the existing experimental design, including the PINK1/Parkin pathway agonist experiments and molecular characteristic analysis, has formed a relatively complete evidence chain. On one hand, the fact that the agonist alone exhibits effects similar to those of AS-IV supports, to a certain extent, the association between the pathway and the drug's action. This is corroborated by Supplementary Figures 2 and 5, as well as relevant literature (PMID: 40490171), which demonstrate the rationality of such a design. On the other hand, this study clarifies the molecular characteristics of AS-IV exerting its effects through activating this pathway, which is a mechanistic complement to our previous work (PMID: 31086091) rather than a mere extension. Regarding your suggestion of supplementing inhibitor or gene knockdown experiments, we fully understand and agree with your emphasis on direct evidence. However, from the perspective of our core goal of elucidating how AS-IV activates the pathway, experiments using agonists with the same direction of action are more in line with the inherent functional characteristics of the drug. Additionally, multiple studies in the field have confirmed the effectiveness of such designs in preliminary mechanistic exploration (e.g., PMID: 39134096). We are concerned that overemphasizing reverse inhibition experiments might, to some degree, shift the focus away from the core finding of the "activation effect". The above is a further elaboration of our research logic. We apologize for any oversights in our previous. We have always held great respect for the rigor of academic research and deeply appreciate that your comments are made out of a sense of responsibility to science. If there is any need for further clarification on specific issues, we will respond promptly and provide relevant information. Thank you again for your time and efforts in reviewing our manuscript. Your opinions are of great significance for improving the quality of our research. |

Reviewer 2 Report
Comments and Suggestions for Authors
The authors have done a good job of revising the article. I believe that the work can be accepted for publication.
Author Response
Dear Reviewer,
Thank you so much for your positive evaluation of our revised manuscript "CIMB-3774335". We are truly encouraged and grateful to hear that you believe the work is suitable for publication.
Your valuable comments throughout the review process have been instrumental in refining our research, helping us strengthen the logic and improve the quality of the manuscript. We greatly appreciate the time and effort you have dedicated to reviewing our work, and your recognition means a lot to us.
We will continue to polish the manuscript carefully in accordance with academic norms to ensure its accuracy and completeness before publication.
Once again, thank you for your generous support and insightful guidance.
Sincerely,
The Authors
Round 3
Reviewer 1 Report
Comments and Suggestions for Authors
The authors are not willing to perform more experiments. So I have no more comments.